# A Comparison of the Catheter-Based Transapical and Surgical Treatment Modalities for Mitral Paravalvular Leak

**DOI:** 10.3390/jcm11174999

**Published:** 2022-08-25

**Authors:** Aleksejus Zorinas, Vilius Janušauskas, Donatas Austys, Giedrius Davidavičius, Lina Puodžiukaitė, Diana Zakarkaitė, Robertas Stasys Samalavičius, Karolis Urbonas, Rita Kramena, Eustaquio Maria Onorato, Kęstutis Ručinskas

**Affiliations:** 1Clinic of Cardiovascular Diseases, Institute of Clinical Medicine, Faculty of Medicine, Vilnius University, Santariskiu 2, LT-08661 Vilnius, Lithuania; 2Department of Public Health, Institute of Health Sciences, Faculty of Medicine, Vilnius University, M.K. Čiurlionio 21/27, LT-03101 Vilnius, Lithuania; 3Clinic of Emergency Medicine, Institute of Clinical Medicine, Faculty of Medicine, Vilnius University, Santariskiu 2, LT-08661 Vilnius, Lithuania; 4Centro Cardiologico Monzino, Istituto di Ricovero e Cura a Carattere Scientifico (IRCCS), University School of Milan, Via C. Parea 4, 20138 Milan, Italy

**Keywords:** mitral paravalvular leak, mitral valve replacement, catheter-based closure

## Abstract

Background: There is a lack of studies where the outcomes of mitral paravalvular leak treatment were compared between surgery and catheter-based closure. The aim of this study was to compare the outcomes of re-do surgery with transapical catheter-based paravalvular leak closure. Methods: This is a retrospective observational study at a single institution; 76 patients were included. According to the treatment, two groups were formed: the “Surgical” group (49 patients after re-do surgery) and the “Catheter” group (27 patients after transapical catheter–based treatment). Results: In-hospital myocardial infarction occurred in 9 (18%) cases in the “Surgical” group and none in the “Catheter” group, *p* = 0.018. Procedure-related life-threatening bleeding occurred in 9 (18%) patients in the “Surgical” group and none in the “Catheter” group, *p* = 0.018. Nine (18%) patients died in 30 days in the “Surgical” group, and none died in the “Catheter” group, *p* = 0.039. A mean follow-up was 3.3 years. No difference was found between the groups by the degree of residual paravalvular regurgitation either at discharge or at follow-up. During the follow-up, 19 (39%) patients died in the “Surgical” group and 2 (7%) among the “Catheter” patients. Conclusions: Transapical catheter-based closure of mitral paravalvular leak seems to be a safer treatment procedure than conventional re-do surgery, and the effectiveness of these procedures does not differ.

## 1. Introduction

Repeat surgery with cardiopulmonary bypass has been the only available effective therapy for the treatment of clinically significant PVL, despite high mortality rates associated with perioperative morbidity [1,2]. The development of catheter-based treatment modalities for structural heart diseases and the need to reduce morbidity and mortality in the treatment of mitral paravalvular leak (PVL) has driven medical professionals along with the medical industry to introduce less invasive treatment—catheter-based PVL closure—into clinical practice [3,4,5]. Undeniably, during the past decade, this treatment option has gained global spread, and in some places, it has become a first-line treatment modality [6,7]. Nevertheless, the comprehensive comparison of long-term outcomes of surgical and catheter-based closure for this PVL is largely unknown, and there is a fundamental lack of data on this issue in the global literature. Few papers exist in which surgical treatment was compared with catheter-based modality treatment for PVL. Due to the recent lack of uniform definitions to determine the significance of the PVL, clinical endpoints to assess safety and efficacy, the authors used MVARC criteria [8,9]. To date, only a few papers exist where “Surgical” treatment compared to the catheter-based modality [10]. Unfortunately, patients in “Catheter” groups are not homogenous, different access sites are employed, and various devices are used for defect occlusion. In this research, we aimed to investigate and compare the results of conventional redo surgery with a homogenous group of patients who underwent transapical catheter-based mitral paravalvular leak closure.

## 2. Materials and Methods

### 2.1. Patient Selection

Vilnius Regional Biomedical Research Ethics Committee and State Data Protection Inspectorate have granted approval for this study (protocol number MVPVF2017). The study received no funding. We retrospectively reviewed all patients who underwent conventional redo surgery or transapical catheter-based procedures for mitral PVL treatment from January 2005 until January 2019. An automatic search of the hospital electronic database for the key word “mitral paravalvular leak” was conducted. Eighty-nine patients were identified in this primary search. We excluded patients with active infective prosthetic endocarditis, dehiscence of prosthesis more than one-third of the annulus perimeter and patients who underwent catheter-based closure with an “off label” device from the formal analysis. Following this refined selection, we remained with a cohort of 76 patients. The group of patients who underwent transapical catheter-based closure of mitral PVL was named “Catheter” and consisted of 27 patients, while the other group of patients who underwent conventional re-do surgery for mitral PVL, named “Surgical,” had 49 patients.

### 2.2. Data Analysis

A comparison of the effectiveness and safety of treatment modalities was performed within the framework of “Clinical Trial Principles and Endpoint Definitions for Paravalvular Leaks in Surgical Prosthesis” [8]. Preoperative clinical and demographic data and general and specific operative variables were investigated. Data were analyzed at baseline, perioperatively, at discharge, at six months and annually after the procedure. Early postoperative characteristics/complications were analyzed at 30 days or in the hospital. Mortality presented as immediate, at 30 days or in hospital and overall at follow-up. The effectiveness of the procedure was evaluated by prosthetic valve function, residual degree of regurgitation at discharge and annual follow-up. The safety was evaluated by the occurrence of morbidity and mortality at the hospital and follow-up. Statistical analysis Statistical analysis was performed using the data collection and analysis software package SPSS 20.0 (IBM Corp., Armonk, NY, USA). The quantitative normality of continuous data was evaluated using the criteria of histograms, rectangular diagrams, and the Shapiro–Wilk test (*p* < 0.05). Quantitative data, distributed as normal, presented as a mean ± standard deviation. The quantitative continuous data distributed outside the normal distribution are presented as the median and quartile intervals. The categorical data are expressed as percentages. Freedom from moderate or severe residual paravalvular regurgitation, new or worsening hemolysis requiring transfusion, new or worsening prosthesis dysfunction and conversion to open surgery, mortality, stroke, readmission for heart failure or treatment of hemolytic anemia were estimated using the Kaplan–Meier method. The censored data included patients who had follow-up terminated. We considered differences statistically significant when the *p* value was lower than 0.05.

### 2.3. Transapical Catheter-Based Mitral PVL Closure Procedure

The procedure was performed in a hybrid operating room (described previously in detail) [11]. In all patients in the “Catheter” group, the PVL closure was achieved using a PLD occluder (Occlutech, Helsingborg, Sweden), which gained the CE mark back in 2014. Prior to skin incision, transthoracic echocardiography is performed to identify the apex of the heart and skin is marked. With a patient in supine position, under general anesthesia and endotracheal intubation, left anterolateral (5–7 cm in length) thoracotomy was performed at a marked location. The pericardium was identified and opened. The procedure is shown in Figure 1. Two pledget reinforced “U” shape sutures were placed and secured with the tourniquets at the apex of the LV (Figure 1A). A needle puncture between pledgets was performed and the guidewire was introduced into the LV, then a short catheter sheath was inserted. Following catheter sheath insertion, the tourniquets gently tightened. A hydrophilic guidewire was used to pass through the defect with the help of a guidance catheter (Figure 1B). The guiding catheter is advanced through the leak, and the hydrophilic guidewire is replaced with stiff wire. The delivery sheath was chosen according to the size of the occluder. The guidance catheter was removed and the delivery sheath advanced through the defect. Under control of real-time 3D TEE and fluoroscopy, a PVL closure device is deployed stepwise, first the distal (atrial) disc. Following the controlled orientation of the device, the distal disc was released from the delivery sheath (Figure 1C). After full expansion of both the proximal and distal occlusion device discs, the function of the prosthetic valve was checked for its interference with the occlude (Figure 1D). If performance of the valve prosthesis is not compromised, position, orientation and hemodynamic effect of the closure device checked (if regurgitation is significantly reduced or not present), the device is detached from the delivery system. Catheters and sheaths were removed from the LV. “U” shape sutures securely tightened and the pericardium closed. Thoracotomy was closed in a routine fashion.

## 3. Results

### 3.1. Preoperative Characteristics

A detailed description is presented in Table 1. A total of 76 patients received mitral PVL treatment from January 2005 until January 2019. Patients in the “Catheter” group were older than in the “Surgical” group, 67 (61–70) versus 64 (57–67) years, *p* = 0.027. The mean perioperative mortality risk according to the European System for Cardiac Operative Risk Evaluation was 6% (4–10%) for the “Catheter” group of patients and 8% (6–11%) for the “Surgical” group. Otherwise, no other differences were found.

### 3.2. Early Postoperative Data and Complications

Early postoperative data and complications in detail are described in Table 2. Significantly higher incidence of myocardial infarction at 30 days after the procedure was among the “Surgical” patients, *p* = 0.01. Life-threatening or disabling bleeding occurred in 9 (18%) patients in the “Surgical” group, and none among the “Catheter” group (*p* = 0.01). The “Surgical” patients statistically significantly lost more blood in the first 24 h after surgery. Due to higher postoperative morbidity, patients in the “Surgical” group spent more time in the intensive therapy unit and in hospital. This high early postoperative morbidity among “Surgical” patients led to higher immediate and in-hospital mortality; none of the patients died in the “Catheter” group at early follow-up.

### 3.3. Results of Mitral PVL Treatment at Discharge from Hospital

Forty patients (82%) out of 49 were discharged alive from the hospital in the “Surgical” group, while all patients went home from the “Catheter” group. With regard to residual paravalvular regurgitation, we found no difference between the groups; data in detail is presented in Table 3.

### 3.4. Results of Mitral PVL Treatment at Follow-Up

Overall, follow-up was available in all discharged patients; the median duration for the “Catheter” group of patients was 2.45 (0.96–3.15) years, while the “Surgical” ones were followed a longer period, 6.3 (2.87–9.3) years (*p* = 0.001). Mortality at follow-up in the “Catheter” group was 8%, and 39% among the “Surgical” cases (*p* = 0.001). The recurrence of significant paravalvular regurgitation of higher than mild degree did not show any statistical significance. Data in detail are shown in Table 4. In addition, Kaplan-Meir survival estimator for composite of death, anemia (Hb < 100 g/L) and residual paravalvular regurgitation higher than mild showed that at follow-up of 2.45 years, freedom from event in the “Catheter” group was 77%, compared to 67% in the “Surgical” group (Log rank, *p* = 0.636, shown in the central image).

### 3.5. Technical Success in the “Catheter” Group

There were no periprocedural strokes. All devices were successfully delivered and positioned, the delivery systems were withdrawn with no complications, and no periprocedural impingement between the device and MVP occurred. There was no immediate conversion to a full sternotomy. Failure to reduce PVL to a mild or lesser degree occurred in one patient; otherwise, technical success was achieved in 26 (96%) cases.

### 3.6. Device Success in the “Catheter” Group

No occluder migration, detachment, fracture, embolization due to thrombosis or endocarditis occurred. Device success was achieved in 23 (85%) patients due to failure to treat severe PVL in one patient, worsening anemia developed in two patients, and one patient had excessive postoperative bleeding requiring surgical revision.

### 3.7. Procedural Success in the “Catheter” Group

All patients were discharged from the hospital. A complete closure of mitral PVL intraoperatively and at discharge (none or trivial residual paravalvular regurgitation) was achieved in 22 (81%) patients, reduction to mild in four (15%) patients; in one patient (4%), the reduction of paravalvular regurgitation was not achieved. The reduction of paravalvular regurgitation to a mild or lesser degree was achieved in 26 (96%) patients. A six-minute walk increased from 264 ± 108 m on admission to 313 ± 120 m (95% confidence interval 20–77 m) (*p* = 0.02) at 30 days after the procedure.

### 3.8. Individual Patient Success in the “Catheter” Group

Individual patient success at one year was achieved in 22 (81%) patients treated. Individual patient success at the one-year follow-up was not achieved in five patients. First is the patient in whom we failed to reduce mitral paravalvular regurgitation (later, this patient expired 12 months after procedure). Another patient died due to uncontrolled sepsis caused by a hemodialysis catheter (the patient was in chronic renal failure preoperatively, which progressed in a few months). Two patients had worsened anemia. Fifth, a patient suffered severe bleeding from a fractured rib.

## 4. Discussion

Access to mitral PVL during catheter-based closure varies depending on the location of the leak or more often on the institution’s chosen preference. A transapical approach in the literature is limited either to case reports or single-center experiences [12]. The evidence in the literature has determined our choice of a surgically open transapical access modality. Some authors have demonstrated a low incidence of adverse procedural events with transapical access sites compared with other access sites for mitral PVL closure; they conclude that the transapical approach could be considered as a first-line therapy [13,14]. Other authors state that this approach allows access to defects in all anatomic locations of the mitral prosthesis [4]. Furthermore, Jelnin et al. showed that a planned transapical approach resulted in shorter fluoroscopy and procedural times compared with converted and combined trans-septal procedures [15].

To date, only a few papers exist where “Surgical” treatment has been compared to the “Catheter” modality and included 848 patients [10]. Unfortunately, none of the publications compares a homogenous group of patients after catheter-based procedure to the conventional re-do surgical group, as in our cohort. In contrast to other authors, our patients were treated through the same access site, and all patients had surgically controlled left thoracotomy for entry into the LV. The defects in our “Catheter” group were closed by a device specifically designed for PVL occlusion. It is also worth mentioning that the “Catheter” group was treated by the same dedicated team of cardiac surgeons, interventional cardiologists, and expert echocardiographic imaging specialists. Some may argue that the “One team” approach may compromise the reproducibility of the procedure. Since mitral PVL complications are relatively rare, to maintain good results, the same team performs its treatment at our center. Thus, if the procedure is performed by various specialists, the procedure results can be compromised by the low volume of performed procedures.

Alkhouli et al. published a comparison of 195 patients who underwent catheter-based treatment for mitral PVL and 186 cases that had redo surgery [16]. In contrast to our group of patients in the catheter-based group, in the Alkhouli et al. group mitral paravalvular defects were approached in three different ways. None of the patients in their cohort was treated in a tranapical approach fashion. What is also worth mentioning is that this group used three different devices, which are “off label” for PVL closure.

Technical success differs between our and Alkhouli et al. groups of the surgical cohorts 90% versus 95.5%, respectively. Comparison of technical success between our catheter-based patients and the Alkhouli et al. group was higher in our group—96% versus 70.1%. Hospital mortality among patients treated surgically was lower among patients in the Alkhouli et al. group compared to our surgically treated patients, 7.7% versus 18%, respectively. In our catheter-based group of patients had a hospital mortality rate of 0%, while it was 3.1% in Alkhouli et al.

Wells et al. compared 58 “Surgical” and 56 “Catheter” patients. Hospital mortality in their “Surgical” arm was 6.9% and in the “Catheter” group—7.1%. Wells et al.’s patients in the catheter-based group stayed shorter at hospital compared to our treated cohort. However, at one year, they found no difference in mortality, readmission or repeat intervention between patients in the “Surgical” and “Catheter” groups [5].

Millán et al. presented outcomes of 163 patients who underwent treatment for mitral PVL surgically or in a catheter-based fashion. In their patients, “Surgical” treatment was applied to 98 patients, and the “Catheter” procedure was performed in 65 cases [9]. The majority of patients—99.3%—treated by redo surgery in their group had no or minimal mitral PVL regurgitation at discharge compared to our surgical patients; this was achieved in 96% of cases. Residual PVL regurgitation higher than mild in the “Catheter” patients of Millán et al. group was 50%, while among our patients treated in the “Catheter” fashion it was 4%. Again, hospital mortality in our “Catheter” group was 0%, while in a publication by Millán et al. it was 2.5%. Comparing redo surgery, in Millán et al., surgical patients’ hospital mortality was 6.6%, while in our surgical patents it was 18%. The remaining two comparative studies consisted of smaller cohorts. Angulo-Llanos et al. reported the results of 67 patients treated for mitral PVL. In patients who underwent “Cathater” treatment, defects were approached in three ways: anterograde (transeptal), retrograde (transaortic) and in a transapical route. Similar to previously presented authors, Angulo-Llanos et al. employed an “off label” device to treat mitral PVL regurgitation. In-hospital mortality among the “Surgical” group of patients in Angulo-Llanos et al.’s publication was 30.6%, compared to our surgical cohort—18% of patients who died in hospital. In contrast, the catheter-based patients in Angulo-Llanos et al.’s cohort had in-hospital mortality at the rate of 9.8%; compared to our patients in the same group, it was 0%. The authors also present their cohort mortality at two-year follow-up, which was 54.3% among surgical patients versus 39.2% in the catheter-based group [17]. In contrast to other authors and our results, Pinheiro et al. presented a smaller cohort, a comparison of 21 patients with mitral PVL; 13 of them underwent redo surgery and eight-catheter-based mitral PVL closure. In their cohort, there were no deaths during hospitalization in the “Catheter” group of patients, while in-hospital mortality among surgical patients was 8%. In addition, it is worth mentioning hospital stay: surgical patients stayed in hospital for 30 days, while catheter-based patients stayed for 32 days [18]. The results of our cohort of patients presented in this manuscript showed that conventional re- do surgery with cardiopulmonary bypass for mitral PVL carries higher early postoperative morbidity, which translates into unacceptably high in-hospital mortality when compared to catheter-based transapical mitral PVL closure with a “purpose specific” device. Similar results are presented in the most respected sources in the literature. In addition, we found that from the perspective of mitral PVL reduction, catheter-based closure of the mitral PVL might be compared with the results of conventional re-do surgery. Nonetheless, the concept of our study is not a randomized controlled trial, but rather a comparison of the retrospectively collected data. This does not allow us to definitively state the superiority or inferiority of either treatment modality with the data available.

### Research Limitations

This research has several limitations. Firstly, this is a retrospective study of a single center practice, where prospectively enrolled patients for mitral PVL treatment in the “Catheter” group compared with a “historical” group of patients who had redo surgery for mitral PVL, when catheter-based treatment for PVL was not available at our institution, this explains the shortness of the follow-up period. Second, the groups compared were not as homogeneous as they could be in a prospective randomized study. Patients who underwent classic cardiac surgery had more invasive and higher-risk procedures. This could be an explanation for the poorer outcomes. Third, the small number of patients aggravates the comparison of the treatment modalities for mitral PVL. Thus, further inclusion of the patients is needed to prove or deny the superiority or inferiority of both methods. In addition, due to the changes in the definitions of the periprocedural myocardial infarction over the time span of inclusion of our patients (2005–2019), we considered the periprocedural myocardial infarction to be significant when the blood serum troponin T was 10 times greater than the normal laboratory values with ischemic ECG changes, hemodynamic compromise necessitating inotropic support and necessity for coronary intervention. Similar issues occurred with the definition of sepsis. It was defined as clinically significant if this diagnosis was present in the patients’ documentation, prolonged intravenous antibiotics usage and the presence of significantly elevated inflammatory blood markers.

## 5. Conclusions

The transapical catheter-based closure of mitral paravalvular leak seems to be a safer treatment procedure than a conventional re-do surgery, and the effectiveness of these procedures does not differ. To definitively support or deny this statement, a randomized controlled trial would be beneficial.

## Figures and Tables

**Figure 1 jcm-11-04999-f001:**
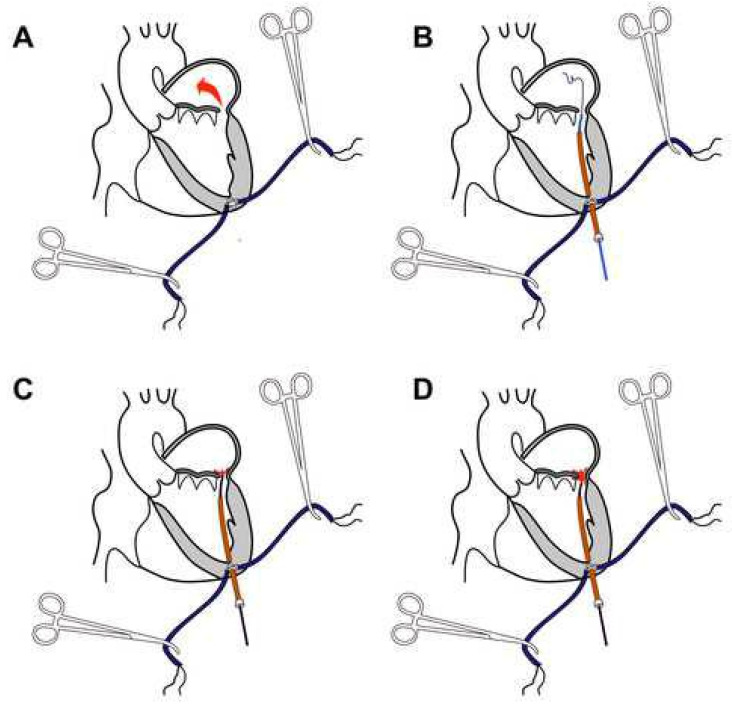
Transapical catheter-based mitral PVL closure procedure: (**A**) “U” shape sutures secured with the tourniquets; the red arrow shows the blood-flow through the defect; (**B**) hydrophilic guidewire passed through the defect; (**C**) release of the distal (atrial) disc of the device; (**D**) release of the proximal (ventricular) disc of the device.

**Table 1 jcm-11-04999-t001:** Preoperative patient characteristics.

Clinical Variables	“Catheter” N (%)/Median [Q1–Q3]	“Surgical” N (%)/Median [Q1–Q3]	*p* Value
Number of patients	27 (%)	49 (%)	
Age, years	67 (61–70)	64 (57–67)	0.027
Gender, male	16 (59%)	22 (45%)	0.231
Time form MVR, months	34 (10–147)	60 (14–179)	0.431
Previous PVL surgery	4 (15%)	5 (10%)	0.552
NYHA			
II	4 (15%)	2 (4%)	0.097
III	21 (78%)	35 (71%)	0.547
IV	2 (7%)	12 (25%)	0.066
EuroSCORE II, %	6 (4–10)	8 (6–11)	0.03
STS risk of mortality, %	2 (1.3–2.6)	2 (1.4–1.2)	0.789
Coronary artery disease	3 (11%)	9 (18%)	0.406
Hemolysis	12 (44%)	15 (31%)	0.228
Anemia Hb < 100g/L	9 (33%)	15 (31%)	0.06
Creatinine concentration, μmol/L	90 (74–107)	88 (77–115)	0.607
Left ventricle function			
Severe (LVEF < 30%)	1 (4%)	2 (4%)	0.935
Moderate (LVEF 31–44%)	10 (37%)	15 (31%)	0.568
Mild LVEF 45–54%)	6 (22%)	22 (45%)	0.05
Good (LVEF ≥ 55%)	10 (37%)	10 (20%)	0.115
PAP > 55 mmHg	12 (56%)	16 (33%)	0.308
Prosthetic valve type			
Bioprosthesis	9 (33%)	4 (8%)	0.005
Mechanical	18 (%)	45 (%)	0.005
Indications for PVL closure			
Hemolytic anemia only	2 (7%)	1 (2%)	0.25
Heart failure only	15 (56%)	34 (69%)	0.228
Both	10 (37%)	14 (%)	0.447
Number of PVL per patient	1 (1–1)	1 (1–1)	
1 defect	14 (52%)	43 (88%)	0.001
2 defects	9 (33%)	5 (10%)	0.013
3 defects	2 (7%)	1 (1%)	0.25
>3 defects	2 (7%)	0 (0%)	0.054
Degree of PVL regurgitation			
Moderate	7 (26%)	17 (35%)	0.431
Severe	20 (74%)	32 (65%)	0.431

EuroSCORE—European System for Cardiac Operative Risk Evaluation; Hb—hemoglobin; LDH—lactate dehydrogenase; LVEF—left ventricular ejection fraction; MVR—mitral valve replacement; NYHA—New York Heart Association heart failure classification system; PAP—pulmonary pressure; PVL—paravalvular leak; STS—The Society of Thoracic Surgery Risk Score.

**Table 2 jcm-11-04999-t002:** Early postoperative data and complications.

Variables	“Catheter” N (%)/Median (Q1–Q3)	“Surgical” N (%)/Median (Q1–Q3)	*p* Value
Number of patients	27 (33%)	49 (67%)	
Immediate mortality (≤72 h)	0 (0%)	4 (8%)	0.127
Mortality (≤30 days/in-hospital)	0 (0%)	9 (18%)	0.039
MI (≤72 h after procedure)	0 (0%)	8 (16%)	0.026
MI (≤30 days or in-hospital)	0 (0%)	9 (18%)	0.018
Stroke (≤30 days or in-hospital)	0 (0%)	3 (6%)	0.19
Bleeding according to BARC			
Life-threatening	0 (0%)	9 (18%)	0.018
Major bleeding	2 (7%)	1 (2%)	0.250
Minor bleeding	1 (4%)	4 (8%)	0.453
Major access site complications	1 (4%)	11 (22%)	0.032
Sepsis	(0%)	7 (14%)	0.039
Drainage, mL/24 h	150 (100–250)	675 (600–1550)	0.001
Hospital stay, days	9 (6–13)	15 (12–21)	0.001
ITU stay, days	1 (1–1)	3 (2–8)	0.001

BARC—Bleeding Academic Research Consortium; ITU—intensive therapy unit; MI—myocardial infarction.

**Table 3 jcm-11-04999-t003:** Results of mitral PVL treatment at discharge.

Clinical Variables	“Catheter PLD” N (%)/Median (Q1–Q3)	“Surgical” N (%)/Median (Q1–Q3)		*p* Value
Patients at discharge	27 (100%)	40	(82%)	0.018
Degree of residual paravalvular regurgitation				
None/Trivial	22 (81%)	36	(90%)	0.316
Mild	4 (15%)	1	(3%)	0.060
Moderate	0 (0%)	1	(3%)	0.408
Severe	1 (4%)	2	(5%)	0.801

**Table 4 jcm-11-04999-t004:** Follow-up results of mitral PVL treatment.

Clinical Variables	“CatheterPLD” N (%)/Median (Q1–Q3)	“Surgical” N (%)/Median (Q1–Q3)	*p* Value
Follow-up available, years	2.45 (0.96–3.15)	6.3 (3.3–10.1)	0.001
Overall mortality	2 (7%)	19 (39%)	0.003
PVL > Mild	2 (7%)	5 (13%)	0.504
Moderate	1 (4%)	1 (3%)	0.408
Severe	1 (4%)	4 (10%)	0.520

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
