# Peer review of "A Comparison of the Catheter-Based Transapical and Surgical Treatment Modalities for Mitral Paravalvular Leak"

_jcm, 2022, doi:10.3390/jcm11174999_

Round 1
Reviewer 1 Report
Zorinas and Colleagues present a study on the treatment of PVL following mitral valve replacement in 76 patients.
1. The title of the manuscript is quite interesting, howerver, the authors do not discuss the role of the pandemic in the treatment of PVL following MVR. This seems to have no correlation to the manuscript.
2. Line 46- 47: This is not comepletely true. The authors have used endpoints quite clearly defined by the MVARC. Please revise.
3. Selection criteria for Catheter based closure is not clear. The EuroSCORE II is significantly lower in the Catheter group. Can the authors explain this ?
4. The authors should name the device used. Did the device have CE certification ?
5. What about patients who underwent mitral valve repair ? Were these patients considered ?
6. Section 2.4. „Conventional redo surgery technique for mitral PVL“, adds no new information and makes the manuscript more bulky.
7. Was 3D-Echocardiography used to guide preinterventional decision making? Can the authors add more detail to the localisation of the PVL ?
8. Did the localisation of the PVL influence the strategy ?
9. Did the presence of biological or mechanical prosthesis influence the procedure ?
10. The Follow-up is quite low (2.45 years ) and in comparision the time span of the study is quite large January 2005-2019, can the authors explain this ? This probably comes from the temporal mismatch ie. comparing a historic group of Re-MVR with a modern Catheter group.
11. Line 213-214: The evidence in the literature has determined our choice of a surgically open transapical access modality. Please elaborate. Did the authors employ other access routes to address the PVL ?
12. Line 279-283: This statement hardly seems justified. The sample size is too low to make such bold statements.
13. The authors end their disscussion with „from the perspective of mitral PVL reduction, catheter-based closure of mitral PVL is not inferior to conventional redo surgery.“ . This seems far-fetched considering the small sample size and the possible temporal mismatch (mentioned in the limitations). This needs to be formulated more carefully.
Minor Revisions:
Line 64: Typo : of
Line 65 : Typo : Device
Author Response
Round 1.
Point 1: The title of the manuscript is quite interesting, howerver, the authors do not discuss the role of the pandemic in the treatment of PVL following MVR. This seems to have no correlation to the manuscript.
Response 1: Thank you for your review. We removed the word “prepandemic” from the title.
Point 2: Line 46- 47: This is not comepletely true. The authors have used endpoints quite clearly defined by the MVARC. Please revise.
Response 2: We revised and updated the introduction.
Point 3: Selection criteria for Catheter based closure is not clear. The EuroSCORE II is significantly lower in the Catheter group. Can the authors explain this ?
Response 3: We excluded patients with active infective prosthetic endo-carditis, dehiscence of prosthesis more than one-third if the annulus perimeter and patients who underwent catheter-based closure with an “off label” devise from the formal analysis.
Point 4: The authors should name the device used. Did the device have CE certification ?
Response 4: We updated the manuscript with this sentence: “In all patients of the “Catheter” group, the PVL closure was achieved using PLD oc-cluder (Occlutech, Helsingborg, Sweden) which has gained the CE mark back in 2014.”
Point 5: What about patients who underwent mitral valve repair ? Were these patients considered ?
Response 5: Rarely, patients following the mitral valve repair do develop PVL. In this analysis, in order to maintain the homogenity of the “Surgical” group we did not include such patients.
Point 6: Section 2.4. „Conventional redo surgery technique for mitral PVL“, adds no new information and makes the manuscript more bulky.
Response 6: We agree with your comment. Therefore we removed the subsection 2.4. “Conventional redo surgery technique for mitral PVL” from the manuscript.
Point 7: Was 3D-Echocardiography used to guide preinterventional decision making? Can the authors add more detail to the localisation of the PVL ?
Response 7: In our practice, we do not consider the localisation of the PVL to be an issue for the transapical closue. The mitral PVL at any location can be easily accessed using this technique. The 3D echocardiography for us is a diagnostic and procedure-guiding tool and does not affect the choice of the treatment modality. It is not used for the preinterventional decision making.
Point 8: Did the localisation of the PVL influence the strategy ?
Response 8: 3D echocardiography does not affect nether the choice of the treatment modality, nether the strategy of the procedure.
Point 9: Did the presence of biological or mechanical prosthesis influence the procedure ?
Response 9: The presence of biological prosthesis did not influence the procedure. The mechanical prosthesis in 2 patients made us to downsize the occluder due to its interference with the prosthesis fuction.
Point 10: The Follow-up is quite low (2.45 years ) and in comparision the time span of the study is quite large January 2005-2019, can the authors explain this ? This probably comes from the temporal mismatch ie. comparing a historic group of Re-MVR with a modern Catheter group.
Response 10: Most of the patients of the “Surgical” group were treated prior the introduction of the catheter-based treatment modality to the practice. At that time, the patients were not considered for the catheter-based treatment due to its absence in the practice. We updated the Limitations section.
Point 11: Line 213-214: The evidence in the literature has determined our choice of a surgically open transapical access modality. Please elaborate. Did the authors employ other access routes to address the PVL ?
Response 11: We describe in detail the literature evidence for the transapical approach: “Some authors have demonstrated low incidence of adverse procedural events with transapical access site compared with other access sites for mitral PVL closure; they conclude that the transapical approach could be considered as a first line therapy [13, 14]. Other authors state that this approach allows access to defects in all anatomic lo-cations of the mitral prosthesis [4]. Furthermore, paper by Jelnin et al. showed that a planned transapical approach resulted in shorter fluoroscopy and procedural times compared with converted and combined trans-septal procedures [15].”
Point 12: Line 279-283: This statement hardly seems justified. The sample size is too low to make such bold statements.
Response 12: We updated the conslusion by softening our statement: “The transapical catheter-based closure of mitral paravalvular leak seems to be safer treatment procedure than a conventional re-do surgery, effectiveness of these procedures does not differ.” Also, in the Limitations section, we state: “further inclusion of the patients is needed to prove or deny the superiority or in-feriority between both methods.”
Point 13: The authors end their disscussion with „from the perspective of mitral PVL reduction, catheter-based closure of mitral PVL is not inferior to conventional redo surgery.“ . This seems far-fetched considering the small sample size and the possible temporal mismatch (mentioned in the limitations). This needs to be formulated more carefully.
Response 13: We softened our statement: “we found that from the perspective of mitral PVL reduction, catheter-based closure of mitral PVL might be compared with the results of conventional re-do surgery”.
Point 14: Line 64: Typo : of
Response 14: Corrected.
Point 15: Line 65 : Typo : Device
Response 15: Corrected.
Reviewer 2 Report
In the manuscript entitled “Prepandemic comparison of the catheter-based transapical and surgical treatment modalities for mitral paravalvular leak” Zorinas et Al. compared outcomes of the mitral paravalvular leak treatment between surgery and catehter-based closure.
The current manuscript deals with an interesting topic of which there are poor data in the literature and this makes it attractive for publication.
It is clearly and properly written, and the overall structure is adequate to the aim; the introduction section provides comprehensive background and appropriately describes the rationale of study. The methodology and results easy to understand, the discussion could be made more fluent. Referencing has been precisely performed.
Some major comments should be addressed
- Compared populations are not homogeneous: patients undergoing classic cardiac surgery are at higher risk and this could explain the increased mortality and the higher rate of complications in this group of patients; sample size should be increased to make groups less differen
- it is not clear what definition was used for post-procedural myocardial infarction; it seems unlikely that no patient undergoing transapical closure had myocardial damage from interventio
- 14% of patients had a sepsis in "Surgical Group" : this data is high compared to those in literature (1); should therefore be explained the definition used for sepsis and detailed better characteristics of patients who developed it.
- It could be explained how methods were used to quantify of residual paralvalvular regurgitatio
BIBLIOGRAPHY
1. Howitt SH, Herring M, Malagon I, McCollum CN, Grant SW. Incidence and outcomes of sepsis after cardiac surgery as defined by the Sepsis-3 guidelines. Br J Anaesth. 2018 Mar;120(3):509-516.
Author Response
Round 1.
Point 1: The discussion could be made more fluent.
Response 1: Thank you for your review. We did some corrections in the discussion.
Point 2: Compared populations are not homogeneous: patients undergoing classic cardiac surgery are at higher risk and this could explain the increased mortality and the higher rate of complications in this group of patients; sample size should be increased to make groups less differen
Response 2: We updated the limitations section: “the groups compared were not as homogeneous as they could be in a prospective randomized study. Patients who underwent classic cardiac surgery had more invasive and higher risk procedure. This could be an explaination for the poorer outcomes.”
Point 3: It is not clear what definition was used for post-procedural myocardial infarction; it seems unlikely that no patient undergoing transapical closure had myocardial damage from intervention
Response 3: We agree with your comment. Due to the changes in the definitions of the periprocedural myocardial infarction over time and the time span of inclusion of our patients (2005-2019) it was difficult to choose the appropriate definition. We also agree that patients in the “Catheter” group had myocardial loss to some extent (apical suture placing), fortunately it did not affect hemodynamics of the patients. Due to all facts mentioned we considered the periprocedural myocardial infarction to be significant, when the blood serum troponin T was 10 times greater than the normal laboratory values with ischemic ECG changes, hemodynamic compromise necessitating inotropic support and necesity for coronary intervention.
Point 4: 14% of patients had a sepsis in "Surgical Group" : this data is high compared to those in literature (1); should therefore be explained the definition used for sepsis and detailed better characteristics of patients who developed it.
Response 4: Due to the prolonged time span of the inclusion of the patients it is difficult for us to determine what definitive criteria of the SEPSIS diagnosis was used a decade or more ago by diferent clinicians at our centre. We considered for it to be clinically significant if the SEPSIS diagnosis was present in the patients documentation and he received prolonged intravenous antibiotics.
Point 5: It could be explained how methods were used to quantify of residual paralvalvular regurgitation
Response 5: We used the 4-class scheme defined by Pibarot P et al. in 2015. The echocardiography was performed by an expert.
Round 2
Reviewer 1 Report
The authors have answered all my querries. I have no further improvements to the manuscript.
Author Response
Round 2.
The authors have answered all my querries. I have no further improvements to the manuscript.
Response 1: Thank you for your review and insights in order to improve the quality of our manuscript.
Reviewer 2 Report
In the revised version of the manuscript entitled "A comparison of the cathether-based transapical and surgical modalitis for mitral paravalvular leak" authors corrected some defects of text form and made it more soundable. The use of heterogeneous definitions for myocardial infarction and sepsis should be included in the study limitations. In the discussion and conclusion should be better expressed the concept that this is not a RCT but a retrospective observational study between heterogenous populations: it's not possibile to define safer the transapical procedure with the avalaible data.
Author Response
Round 2.
Point 1: The use of heterogeneous definitions for myocardial infarction and sepsis should be included in the study limitations.
Response 1: Thank you for your review. We updated the limitations section.
Point 2: In the discussion and conclusion should be better expressed the concept that this is not a RCT but a retrospective observational study between heterogenous populations: it's not possibile to define safer the transapical procedure with the avalaible data.
Response 2: We modified the discussion and conclusions in accordance with your remarks.